# How minimizing conflicts could lead to polarization on social media: An agent-based model investigation

**Michele Coscia***, **Luca Rossi**

IT University of Copenhagen, Copenhagen, Denmark

* mcos@itu.dk

## Abstract

Social media represent an important source of news for many users. They are, however, affected by misinformation and they might be playing a role in the growth of political polarization. In this paper, we create an agent based model to investigate how policing content and backlash on social media (i.e. conflict) can lead to an increase in polarization for both users and news sources. Our model is an advancement over previously proposed models because it allows us to study the polarization of both users and news sources, the evolution of the audience connections between users and sources, and it makes more realistic assumptions about the starting conditions of the system. We find that the tendency of users and sources to avoid policing, backlash and conflict in general can increase polarization online. Specifically polarization comes from the ease of sharing political posts, intolerance for opposing points of view causing backlash and policing, and volatility in changing one's opinion when faced with new information. On the other hand, it seems that the integrity of a news source in trying to resist the backlash and policing has little effect.

## 1 Introduction

Many people use social media as their primary channel for news consumption [1]. However, there is a growing concerns about information quality and truthfulness to be found there [2–4]. This has prompted researchers to investigate the effects on society of low-quality online information offer. Results point out to negative effects from social media: increase in ideological segregation [5, 6], in polarization [7], in distrust towards mainstream media [8, 9], and amplified effects of selective exposure [10]. Many of these social dynamics are deeply intertwined and, when supported or reinforced by social media platforms' algorithmic sorting, produce a scenario where users' belief in certain narratives has little to no connection with the factual nature of the narrative itself [10]. A consolidated approach to deal with the problem of growing low-quality information is based on users flagging content that they deem to be unreliable or problematic. Nevertheless, in a previous study, authors highlighted how traditional flagging systems penalize neutral and factual news sources by employing naive assumptions on how users flag content online [11].

**Data Availability Statement:** All relevant data are within the paper and its Supporting information files.

**Funding:** The author(s) received no specific funding for this work.

**Competing interests:** The authors have declared that no competing interests exist.

In this paper, we want to investigate the link between how users and news sources interact on social media and their polarization. Specifically: under which circumstances of tolerance, ease to share content, news source integrity, and user opinion volatility do we see an increase or a decrease in political polarization?

One issue with this type of research question is that it requires the quantification of processes happening inside a user's head or a news source's editorial board. These are usually hard to measure quantities and there is little agreement of how they should be estimated. One promising way to investigate such questions is via the definition of a realistic agent-based model. Agent-based models proved their usefulness in many scenarios involving questions difficult to test empirically or non-amenable to closed form solutions. In fact, there are multiple examples of usage of agent-based models exactly to study online news consumption and polarization [6, 10, 12–19].

The general approach behind the cited papers is similar. First, social relations among actors carrying different opinions are modelled on a network. Then different models of social influence and network evolution (e.g. unfriending, rewiring) are developed to observe under which values polarized communities emerge. Within this vast body of research our main point of reference is the post transmission-distribution model [20]. Just like in [20], we go beyond most of the previous literature by creating a realistic model in which users can have a continuous opinion value that can change over time, by being attracted towards similar points of view or repulsed by different points of view.

We expand over [20] and over the existing research [6, 10] in several ways. First, differently from [20], we model polarization as a bipartite network with two types of actors: users *and* sources. This allows us to study a set of more complex dynamics such as polarization of both sources and users (according to their own internal processes) as well as how the polarization of one type of actor can affect the other.

Second, thanks to the bipartite approach we adopt, we can study how polarization dynamics impact the audience networks and how that can shape the behaviour and the strategy of news organizations.

Third, compared to existing research [20, 21] we have more realistic starting conditions of user opinions and of the underlying networks. In our model, the opinions are not distributed uniformly at random, but each user's opinion is correlated with their neighbors' opinions, as homophily is a well-documented property of online social networks [22]. Moreover we adopt, for generating the social network connecting users, a Lancichinetti–Fortunato–Radicchi (LFR) benchmark [23]. This creates a social network with a realistic community structure, a broad degree distribution, and overlapping communities. All these characteristics are typical of real-world social networks.

We find that the increase of polarization in both users and sources is supported by a wide range of scenarios. Specifically, we find direct positive associations between polarity increase and: the amount of content shared online, the intolerance of users in regard to content far from their opinion, and the volatility of users in changing their opinion based on what they read online. The integrity of news sources, i.e. how much they stick to their own polarization rather than avoiding backlash, plays a negligible role in increasing or decreasing polarization.

In practice, both users and sources in our model try to minimize conflict, by changing their opinions towards the ones minimizing backlash and/or removing relationships to friends/news agencies that are too far to be reconciled. Yet, this conflict avoidance often results in a polarized environment.

Our results should be taken with caution, due to the fact that they are based on the assumptions baked in our model and on synthetic data. Moreover, our parameters interact with each other in non trivial ways. For instance, user opinion volatility increases polarization for low values of tolerance, but it decreases polarization for high values of tolerance.

Nevertheless, given the complexity of the topic, we think that these results provide interesting insights that should be used as the basis for further empirical experiments on the effect of social media consumption of news.

The archive containing the data and code necessary for the replication of our results can be found at https://www.dropbox.com/s/rldphdm8w6letox/20211020_flagging_code.zip?dl=0.

## 2 Motivation

In the original paper [11] we showed how a more dynamic understanding of users' behaviour and of the impact that users' social networks have on news exposure can explain some unexpected results observed in real world social media flagging data. We now extend it to describe more complex issues of online users' polarization, and to incorporate an even larger number of social dynamics. Both these issues and dynamics are based on observations in empirical research.

As detailed in the next section, we assume a model with two types of actors: users and media sources. Users receive news items shared by media sources and are able to re-share those within their social network or to *flag* them if they deem them unacceptable. This basic principle of the model is unchanged compared with the original paper [11] and its underlying principles have already been justified with both empirical data and existing research.

This second iteration of the model adds the ability for users and media sources to evolve depending on other actors' activities. More precisely actors can—in addition to consume, share, and re-share news content –: change their political idea (being affected by the opinions of their social networks and the content they are exposed to), rescind ties with friends (unfriend) and media sources (unfollow) whenever their polarity is too far from the user's own polarity.

Media sources can—in addition to share news-items on the network—adjust their polarity (and thus the polarity of the news items they share) based on the average polarity of their audience. Sources *follow* the audience towards a higher or lower level of polarization.

Before we detail the construction of the model, we need to explain the social dynamics we want to model and the empirical evidences supporting this decision.

### 2.1 Users' behaviour

The effect of social media exposure on individuals is arguably one of the most studied aspects of digital platforms. There is a growing amount of evidence pointing at the dynamic nature of human behaviours when it comes to dealing with their online experience. This, we argue, operates following two principles: a) users are affected by social influence, b) users are active actors in shaping their online connections. These principles have been successfully included in previous models dealing with users' polarization [18, 19].

Most studies, especially those built on large datasets and quasi-experimental settings, agree that messages shared on social media directly influenced political self-expression, information seeking and real-world voting behaviour [24]. Nevertheless, the way in which these effects materialize is far from simple. On the one side they are mediated by platforms' affordances [25] meaning that we will observe different effects (or different magnitudes of the same effect) on different social media platforms, and, on the other side, they interplay with pre-existing individual predispositions. This is shown in field experiments [26]. Exposure to political messages not always produces a change in political polarity aligned with that of the message but that it often has a repulsive effect. In the context of [26], Republican voters reported substantially more conservative views after being exposed to liberal messages, while the opposite effect was not observed in a significant way.

In addition to changing their opinion according or in opposition to what they see, users can also decide to rescind their online connections in order to minimize their exposition to unwanted or annoying content. This behaviour has been extensively studied in the literature [27–29] and it contributes to a constant evolution of the actual social network structure that could potentially lead to higher level of polarization and echo-chamber effects. Research seems to agree that actual removal of connections that are not aligned with the users' opinions is a relatively rare but existing phenomenon [27, 28, 30] but the actual mechanisms behind it are still uncertain and somehow contradicting results have been reported: [28, 29] report, using survey data, how unfriending practices were more common among those who had a more political use of social media and a higher level of engagement. Nevertheless, these results were not confirmed by [31] that reported largely opposite results noting how neither engagement nor level of disagreement were found to be significantly related to filtering out of contacts.

## 2.2 Media sources' behaviour

Media sources, as well as users, need to be represented in a less static way. The contemporary media economy presents an unprecedented situation for media actors that appears to be, at the very same time, dependent on social media platforms to reach their readers while often lamenting that their work is freely used, without any licensing fee, by the same social media giants [32]. The changes in the dissemination platforms as well as in the revenue model of news-media actors have been connected both with the emergence of click-baiting [33] as well as to increased level of partisanship that has, in recent times, given a more and more central role to hyperpartisan news actors [34–36] Within the described context, news-media actors, while still trying to avoid the backlash from the public, are encouraged to cultivate their own partisan communities [37] that, through their networked connection, will facilitate news-media's long-term existence.

## 2.3 On generalizability and validation

We propose a general model accounting for user-user and user-source dynamics. While this is not modelled after any real-world online social network it has characteristics that can be founded on many online platforms and it can be used to simulate the underlying social dynamics that involve both users and news-sources. While data to perform a precise validation of the proposed bipartite model is not currently available, we validated the model's ability to produce realistic results using real-world data from Twitter as described in Section 4. More precisely we validated the ability of the model to produce a network of users as polarized as the one we observed on the validation data collected from Twitter. We did this following the procedure described in [21]: we collected the same twitter data used by [21] on the polarizing issues of abortion in the US and used it to build a user-user network with known users' leaning. Using this network we calculated the Pearson correlation between the distribution of the polarity scores on the Twitter data and what we could achieve with our model. The results described in Section 4 show the ability of the model to produce realistic results in a parameter space and confirm and integrate the observation from other research validated on real-world social media data [10, 18, 19].

## 3 Model

The previous section provides the theoretical ground for all elements of the model, relying on established studies in the literature, specifically [11]. In this section, first we give a general intuition of the model. Then, we formally define all of its parts:

- The agents: users and news sources;

- The structures connecting the agents;

- The actions the agents can perform;

- The phases of the model, putting together all the parts of the model into an iterative process.

### 3.1 Intuition

News sources publish news items with a polarity value. The user audience of each source reads the news items published by the source. Users flag the content they disagree with and reshare the content they agree with. Resharing means that all their friends in the social network will also see the content. They can flag it, or reshare it, thus generating a cascade.

Users update their polarity by being attracted by friends and content that is similar to them, and by being repulsed by friends and content that is dissimilar. Sources update their polarity by testing whether moving to their audience's average polarity will result in fewer flags.

Users remove their connections to friends and sources with different polarity than their own. Users make new friends among those who follow similar news sources. Users will follow new news sources with a polarity similar to their own.

### 3.2 Agents

The two agent types are news sources and users.

A news source $s$ has polarity $p_{s,0}$ at initiation $t = 0$, which is normally distributed between $-1$ and $+1$: most sources are neutral ($p_{s,0} \sim 0$) and there are progressively fewer and fewer sources that are more polarized ($p_{s,0} \sim -1$ or $p_{s,0} \sim +1$). See Fig 1(a) for a reference. Each news item $i$ published by a news source at time $t$ carries its polarity value, $p_i = p_{s,t}$.

A user $u$ has an initial polarity $p_{u,0}$. It distributes the same way as that of the news sources: most users are moderate and extremists are progressively more rare. See Fig 1(b) for a reference.

Our assumption of a normal distribution for the initial polarity is supported in Section 1 in S1 File. Overall, we work with a system containing 800 sources and 942 users.

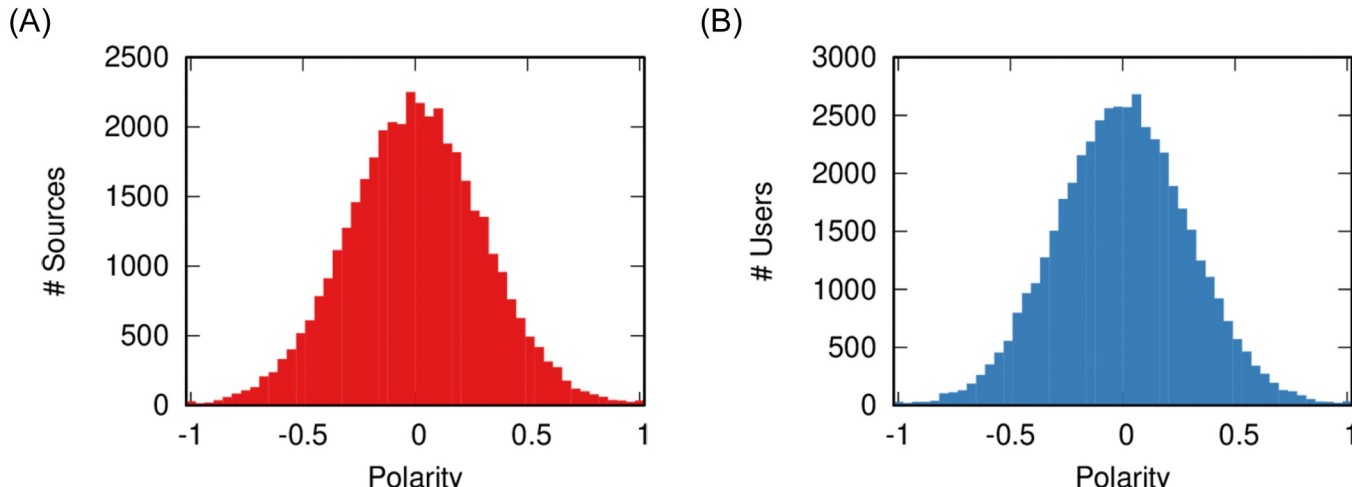

**Fig 1. The initial polarity distributions for (a) sources (in red) and (b) users (in blue).** The plots report how many sources/users (y axis) have a given polarity value (x axis).

### 3.3 Structures

Users connect to the news sources they follow in a bipartite network. We call it the audience network. Users tend to connect with the sources with the most similar polarity possible. News sources have a broadly distributed degree in this network, which is obtained from real world data—which we extracted from CrowdTangle: https://www.crowdtangle.com/. Fig 2(a) shows an example of initial audience network. $A$ is an $s \times u$ bipartite matrix. Notation-wise, $A_{u,t}$ tells us the set of sources $u$ follows at time $t$, while $A_{s,t}$ tells us the set of users following $s$ at time $t$.

Users also connect to each other in a social network. The social network has many realistic topological characteristics: broad degree distribution, clustering, and communities. We generate it using the LFR benchmark [23]. Users in the same community tend to have the same polarity. Specifically, we iterate over all communities generated by the LFR benchmark and we assign to users grouped in the same community a contiguous portion of the polarity distribution. Fig 2(b) shows an example of initial social network. We use $N_{u,t}$ to refer to the set of users connected to $u$ at time $t$.

The number of connections in the audience and social network depends on a randomization process that produces different networks from run to run. A sample run has 8, 086 edges in the social network (average degree of 17) and 6, 123 edges in the audience network (average degree of 7).

### 3.4 Actions

**3.4.1 Users.** Users can: reshare, flag, change polarity, unfriend, and unfollow.

*3.4.1.1 Resharing & flagging.* Every time user $u$ at time $t$ sees a news item $i$, it calculates the difference in polarity as $|p_{u,t} - p_i|$.

If $|p_{u,t} - p_i| < \rho$, meaning that the polarity difference between user and item is sufficiently low, then $u$ reshares $i$, meaning that all of $u$'s friends will also see $i$. Thus, $\rho$ can be interpreted as the "propensity to reshare": low $\rho$ equals to small cascades.

If $|p_{u,t} - p_i| > \phi$, meaning that the polarity difference between user and item is sufficiently high, then $u$ flags $i$. Thus, $\phi$ can be interpreted as the "tolerance" of users: low $\phi$ equals to more flags.

If neither is true ($\rho \leq |p_{u,t} - p_i| \leq \phi$), the user will do nothing.

*3.4.1.2 Change polarity.* At every time step $t$, user $u$ updates its polarity. This depends on the polarity of their direct friends and of the news items they read. We start by considering the social network.

A friend $u'$ of $u$ with polarity similar to $u$ will attract $u$ to that polarity—i.e. the friends are pulling each other. Vice versa, if $u'$ is far from $u$'s polarity, then the friends will repel (or push) each other. This is also regulated by the tolerance parameter $\phi$. The set of pulling friends is $N_{u,t}^l = \{u' \mid |p_{u,t} - p_{u',t}| \leq \phi, u' \in N_{u,t}\}$ and the set of pushing friends is $N_{u,t}^s = \{u' \mid |p_{u,t} - p_{u',t}| > \phi, u' \in N_{u,t}\}$.

The pull experienced by $u$ from $u'$ is $(p_{u',t} - p_{u,t})$. We can aggregate the collective pull of all $u'$ friends as $d_{u,t}^l = \overline{p_{u',t}^l} - p_{u,t}$, with

$$\overline{p_{u',t}^l} = \frac{\sum\limits_{u' \in N_{u,t}^l} p_{u',t}}{|N_{u,t}^l|}.$$

The push strength is calculated in the exact same way, but reversing the polarity difference and replacing $\overline{p_{u',t}^l}$ with $\overline{p_{u',t}^s}$: $d_{u,t}^s = p_{u,t} - \overline{p_{u',t}^s}$.

(A)

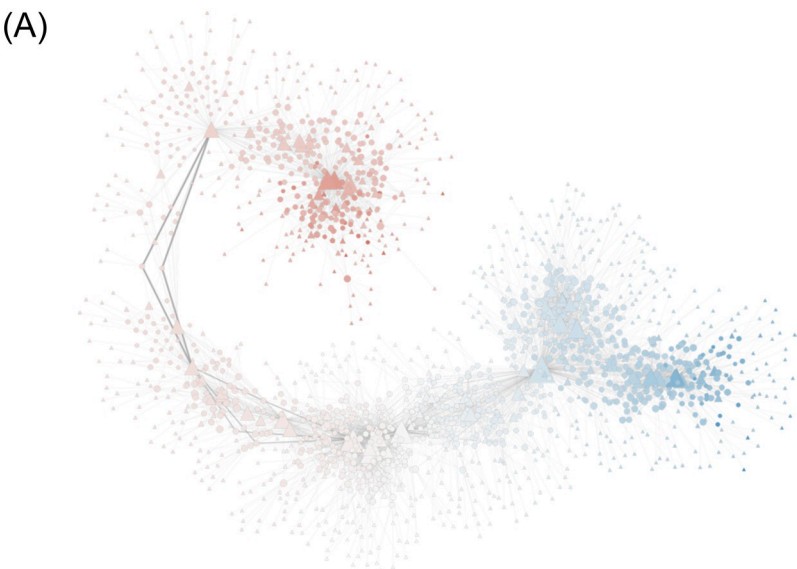

(B)

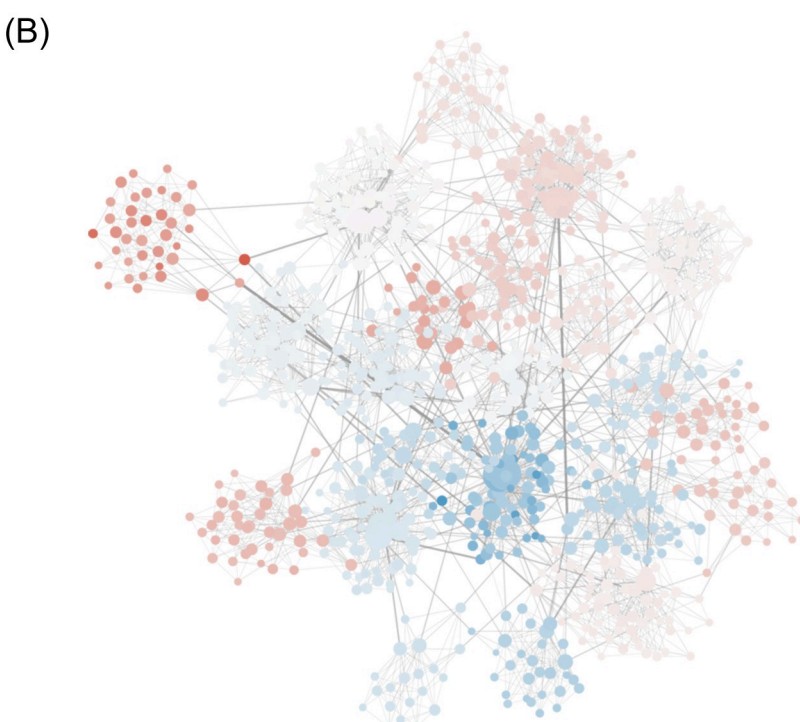

**Fig 2. The initial structure configuration for the (a) audience and (b) social networks.** Node color is proportional to their polarity (from blue equal to −1 to red equal to +1, passing via white equal to 0).
In the audience network, sources are triangles, while users are circles in both figures. Node size is proportional to the degree. Edge size and transparency is proportional to edge betweenness.

The total pull experienced by $u$ is the sum of pull and push, weighted by the number of friends pulling and pushing:

$$d_{u,t} = \left(\frac{\sigma}{|N_{u,t}|}\right)((|N_{u,t}^l|d_{u,t}^l) + (|N_{u,t}^s|d_{u,t}^s)).$$

The parameter $\sigma$ regulates how "volatile" users are: if it is 0 users never update their polarity, with $\sigma = 1$ they update their opinion by weighting this pressure as much as their previous opinion.

Formally, $p_{u,t} = p_{u,t-1} + d_{u,t-1}$.

The news items a user $u$ reads also change $u$'s polarity. The formula is the exact same as the one above, with low-polarity-difference items pulling and high-polarity-difference items pushing. Note that the pull/push from the news is calculated considering all items $i$ a user $u$ sees, regardless whether $u$ follows the news sources directly or the items were shared by $u$'s friends.

As the result of this process, it is possible that some $p_{u,t}$ and $p_{s,t}$ values will be higher than +1 or lower than −1. In these cases, we clip these values so that the boundaries of the polarity space are respected.

*3.4.1.3 Unfriend & unfollow.* Users break all their connections—with both friends in the social network and sources in the audience network—if the polarity difference is larger than the tolerance parameter $\phi$, meaning that $N_{u,t} = N_{u,t-1} - N_{u,t-1}^s$ and $A_{u,t} = A_{u,t-1} - A_{u,t-1}^s$.

Each user will then try to create new connections in both networks. For the social network, they will pick new friends with a probability proportional with the number of common news sources the users are following. The user similarity is calculated as $A^T A$. For the audience network, they will pick new sources with a probability proportional to their polarity similarity— i.e. the similarity is $|p_{u,t} - p_{s,t}|$. If the user picks a source that they were already following, this will increase the audience edge's weight by one.

**3.4.2 News sources.** In the model, a news source $s$ performs two actions. First, it publishes a news item. This means that all users connected to $s$—its audience—will see and react to it.

Second, at time $t$, $s$ will update its polarity $p_{s,t}$. It does so with a two-phase move. First, it calculates the average polarity of its audience:

$$\overline{p_{s,t}} = \frac{\displaystyle\sum_{u \in A_{s,t}} p_{u,t}}{|A_{s,t}|}.$$

Then it simulates a cascade: it publishes a simulated news item $i$ with polarity $\overline{p_{s,t}}$ and it records the hypothetical number of flags it would receive in total.

The number of flags $s$ receives at time $t$ with its polarity $p_{s,t}$ is $f_{s,t}$. The number of flags $s$ would have received with polarity $\overline{p_{s,t}}$ is instead $\overline{f_{s,t}}$. Their relative difference, $F_{s,t}$ is an important quantity:

$$F_{s,t} = \frac{f_{s,t} - \overline{f_{s,t}}}{f_{s,t} + 1}.$$

The plus one in the denominator solves the issue of sources receiving zero flags. If $F_{s,t} > 0$, then $s$ would receive fewer flags if it were to change its polarity. Thus $s$ is incentivized to switch its polarity. To sum up:

$$p_{s,t} = \begin{cases} \overline{p_{s,t-1}} & \text{if } F_{s,t-1} > \gamma \\ p_{s,t-1} & \text{otherwise.} \end{cases}$$

Here, $\gamma$ is an "integrity" parameter: higher values of $\gamma$ mean that the source will attempt to keep its polarity and resist more pressure. It will only concede if the reduction in number of flags is significant.

### 3.5 Action phases

The model starts with users and sources connecting in an audience and social network and carrying a given polarity (Fig 3(a)). At each time step $t$, the model runs through five phases:

1. **Cascade**: the news sources publish their items (Fig 3(b)), which are reshared and flagged by the users (Fig 3(c)), regulated by the $\rho$ and $\phi$ parameters. Note how in the figure user $U4$ flags source $S1$ even if they do not follow it, because they read its content via the share from user $U1$.

2. **Update user polarity** (Fig 3(d)): the users update their polarity according to the pull and push they experience from their friends and all the news items they read, regulated by the $\phi$ and $\sigma$ parameters. Note how $U1$ becomes more extreme red even if $S1$ and $U2$ would pull it

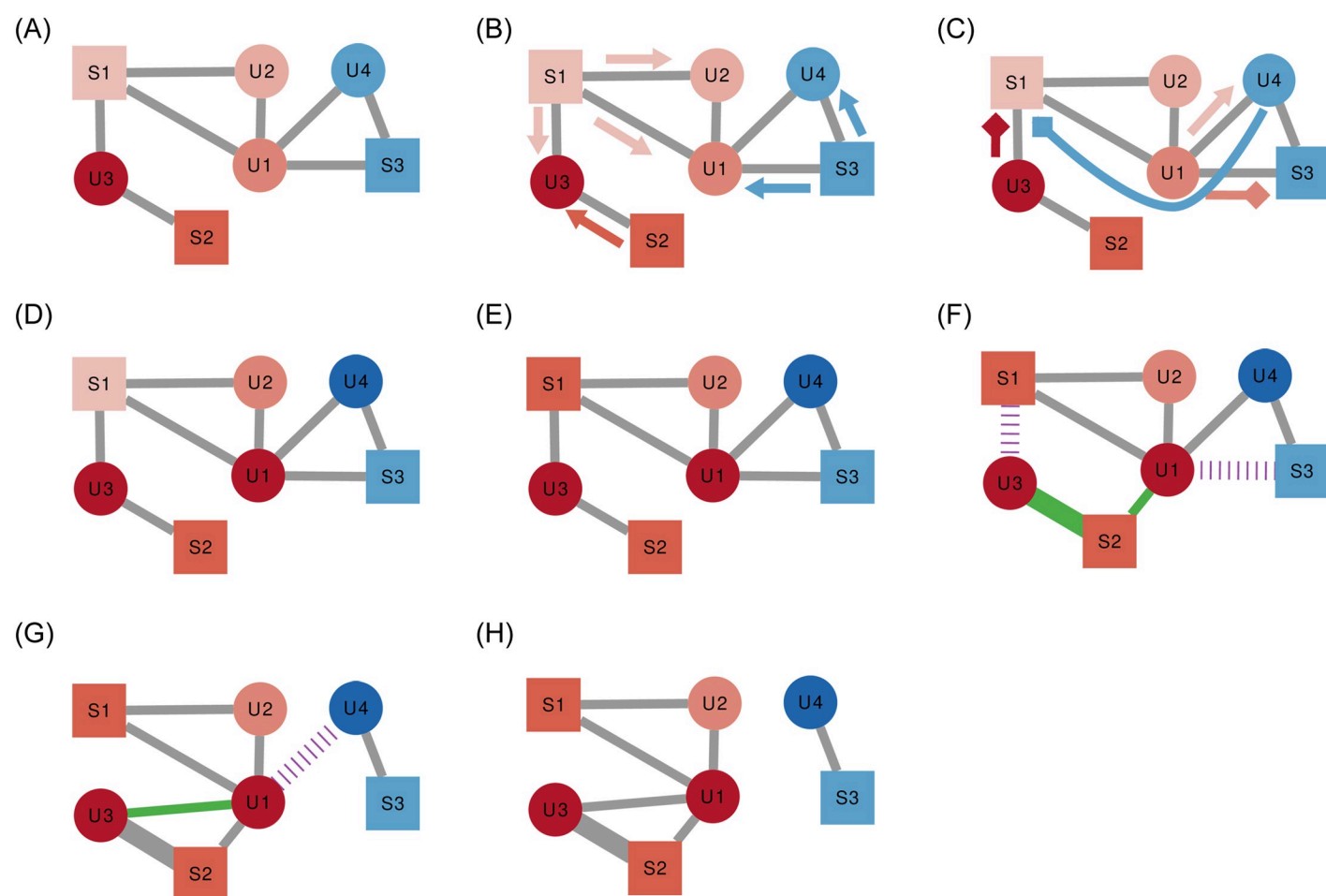

**Fig 3. The steps of the model.** The shape of the nodes indicates the type (square = news source, circle = user). Colors encode polarity (red = positive, blue = negative). Arrows show news items (triangular head) and flags (square head). Deleted connections in purple slashes, new connections (or connections with increased weight) in green. (a) Initial condition. (b) Sources publish their items. (c) Users reshare and flag. (d) Update user polarity. (e) Update source polarity. (f) Rewire audience network. (g) Rewire social network. (h) Step completed.

to moderation. The reason is the strong push *U*1 feels from *U*4 and *S*3. Similarly, *U*4 is pushed by *U*1 and *S*1.

3. **Update source polarity** (Fig 3(e)): the sources update their polarity to the average polarity of their direct audience, but only if that would result in a lower number of flags, regulated by the γ parameter. *S*1 received two flags so it averages *U*1, *U*2, and *U*3's *initial* polarity (from Fig 3(a)). *S*2 does not move as it did not receive any flag. *S*3 does not move because, even if it were to move, it would still receive a flag from *U*1.

4. **Update audience network** (Fig 3(f)): the users unfollow sources they flagged (if they were following them) and redirect their attention to sources with a similar polarity, regulated by the ϕ parameter. Note how *U*3 has no choice but following *S*2 with twice the strength as before, to make up for unfollowing *S*1.

5. **Update social network** (Fig 3(g)): each user unfriends users with a too high polarity difference and befriends people following similar news sources, regulated by the ϕ parameter. *U*1 had both *S*1 in common with *U*3 (again the information used comes from Fig 3(a)), thus that is the most likely new link to be created.

Fig 3(a) is the configuration of the system at time $t - 1$ and Fig 3(h) is its result once time step $t$ is complete. Note how the network broke into disconnected components and how the average polarity (represented by the hue intensity) has increased.

## 4 Validation

To investigate the behavior of the system for different parameter values, we make a grid test of all four parameters in the value intervals that show the maximum variation.

Table 1 reports a summary of all the tested values per parameter. Outside the tested parameter values we do not observe significant changes—see Section 2 in S1 File for more details. Section 3 in S1 File provides empirical data as evidence for the model reaching convergence before the 100th step, which is when we terminate it.

To validate our model we collect data from Twitter. Specifically, we build a follower-followee network and estimate a user's polarity by averaging the polarity of the news source URLs they share. This is conceptually aligned with what introduced by [38] and follows the procedure used in [21], as well as using the tweet IDs they shared on the topic of abortion.

This results in a network with a given polarity distribution for the users. Note that we have to re-normalize the real world polarity distribution, because our model allows for extreme +1 and −1 scores, while the data source estimating the source polarization does not allow such scores.

We bin the user polarity distribution, counting how many users are part of each polarity value bin with 0.1 increments. We do so both for the real distribution and for each distribution we obtain for each parameter combination. Then, we calculate the Pearson correlation of the

**Table 1. The full set of parameter values for all parameters of the model that we test in the paper.** We have tested all combinations of these parameters, although some results are omitted for clarity. (ρ: Reshareability; ϕ: Tolerance; γ: Integrity; σ: Volatility).

| ρ | ϕ | γ | σ |
|------|-----|-----|------|
| 0.06 | 0.1 | 0.0 | 0.1 |
| 0.07 | 0.2 | 0.1 | 0.5 |
| 0.08 | 0.3 | 0.2 | 0.66 |
| 0.1 | 0.5 | 0.4 | 1.0 |

**Table 2. The Pearson correlation values for the best (top) and worst (bottom) parameter configurations in the model (higher value is better).**

| $\rho$ | $\phi$ | $\gamma$ | $\sigma$ | Correl |
|---|---|---|---|---|
| 0.1 | 0.1 | 0.2 | 1.0 | 0.7170 |
| 0.1 | 0.1 | 0.4 | 1.0 | 0.7146 |
| 0.1 | 0.1 | 0.1 | 1.0 | 0.7144 |
| 0.08 | 0.1 | 0.0 | 1.0 | 0.7138 |
| 0.08 | 0.1 | 0.2 | 1.0 | 0.7125 |
| . . . | . . . | . . . | . . . | . . . |
| 0.1 | 0.5 | 0.1 | 1.0 | -0.4868 |
| 0.06 | 0.2 | 0.0 | 0.5 | -0.4879 |
| 0.06 | 0.5 | 0.0 | 1.0 | -0.4897 |
| 0.08 | 0.5 | 0.0 | 1.0 | -0.4916 |
| 0.06 | 0.2 | 0.4 | 0.5 | -0.4924 |

counts in each bin. A high Pearson correlation means that there are relatively as many users in each bin in the real distribution as well as in the simulation.

Table 2 shows the five best and worst performing combinations of parameters according to the Pearson correlation. We can see that the model can achieve a good alignment with real world data, as there are configurations of parameters that show $> 0.7$ correlations ($p \sim 0.0003$). Some parameter configurations are led astray, with negative correlations. However, we can see that there is a parameter space where the model returns reasonably realistic outputs.

If we aggregate results by parameter, calculating the average correlation for each of them, we get the best performance with high shareability, low tolerance, and high volatility. Integrity seems not to matter too much.

# 5 Results

We discuss the results by exploring the effect of the parameters $\rho$ (the propensity of users to reshare), $\phi$ (the tolerance of users in flagging), $\gamma$ (the integrity of the sources in maintaining their polarity and resisting the signals from received flags), and $\sigma$ (the volatility of users).

In all the subsequent sections we show three things: the distributions of the source (red) and user (blue) polarity; and the social network resulting from the rewiring. All results (except the social network snapshot) are the aggregated result of 50 independent runs.

Note that these results are robust to small variations of the model. Specifically, in Section 4 in S1 File we test what happens when sources do not only try to minimize the backlash against them, but also try to maximize the spread of their news items.

To aid intuition, in Section 5 in S1 File we provide a run through one instance of one parameter combination.

## 5.1 The effects of sharing ($\rho$)

We first test how sources react to environments where there is a different propensity by the users to share stories. For low values of $\rho$, users only share news that are very close to their world views; conversely high $\rho$ means that there is more sharing.

No matter the $\rho$ value, both users and sources experience a similar dynamic (Fig 4(a)–4(f)). Three clusters of polarity emerge: the moderates in the middle and the extremists at both ends of the polarity values.

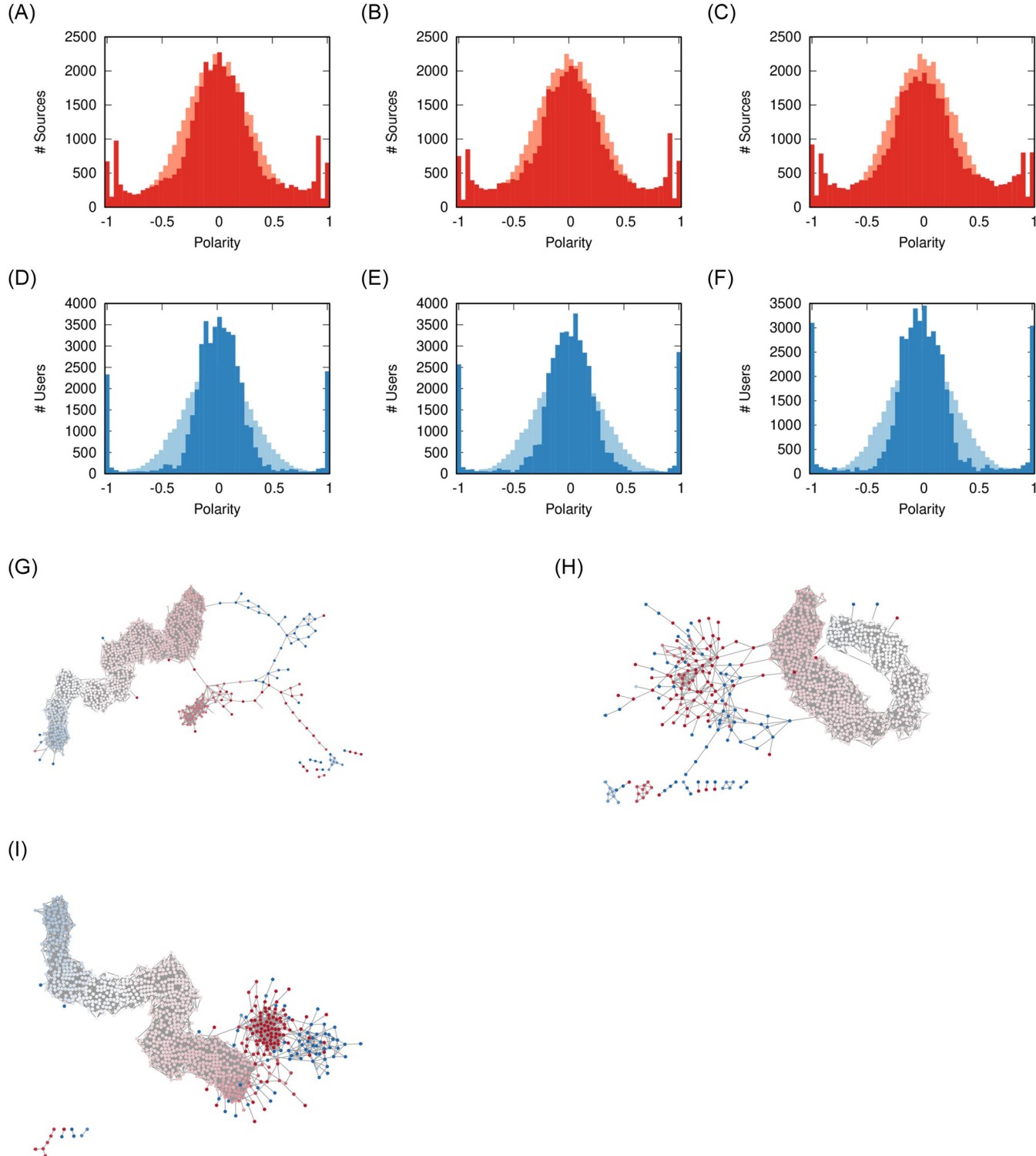

**Fig 4. Effect of $\rho$ on user (a-c) and source (d-f) polarity distributions, and (g-i) on the social network.** The pale distribution in the background is the original one. Other parameters fixed at: $\phi = 0.1$, $\gamma = 0$, $\sigma = 2/3$. The nodes in the network are colored proportionally to their polarity from −1 (blue) to +1 (red) passing via 0 (white). (a) $\rho = 0.06$ (b) $\rho = 0.07$ (c) $\rho = 0.08$ (d) $\rho = 0.06$ (e) $\rho = 0.07$ (f) $\rho = 0.08$ (g) $\rho = 0.06$ (h) $\rho = 0.07$ (i) $\rho = 0.08$.

While moderate sources still exist, higher values of $\rho$ create more polarization in the users: compare the left-right peaks in Fig 4(d) and 4(f). This is due to the fact that, when there is more sharing, users are more likely to read items carrying a larger polarity difference and thus they are pushed more. This is supported by the final social network configuration: in Fig 4(g)–4(i) we see that more sharing creates larger and larger extremist clusters (solid blue and red). The clusters get isolated from the main core of the network, which organizes itself on a polarity gradient.

Social media facilitate sharing, thus promoting higher values of $\rho$. This is supported by our validation, showing most realistic results for high $\rho$. Our results from this section suggest that this might increase polarization, because it increases chances of conflict and thus pushes users and sources more to avoid them. A lower amount of sharing would limit the size of the polarized clusters.

## 5.2 The effects of tolerance ($\phi$)

We now turn to how sources react to environments where there is a different propensity by the users to flag stories. For low values of $\phi$, users flag more, because even news at a low $\phi$ distance from their polarity could be flagged. Conversely, high $\phi$ means that there is high tolerance.

Fig 5(a)–5(f) shows that, for low values of $\phi$ ($\phi$ = 0.1, more flagging), most articles percolating through the social network are flagged by the users. The emergence of the three polarity clusters (−1, 0, and + 1) force sources to adapt. When we increase the tolerance of users, rather than polarizing, the users and sources instead tend to converge more to the middle.

Looking at the social network shows that, in a high tolerant society (Fig 5(i)), we do not see the erosion of communities that we see in all other cases. That is because a high $\phi$ means minimal unfriend events, which only involve the inter-community links between communities with different polarity.

Previous results [11] suggest that users are not tolerant, as low $\phi$ values tend to better reproduce real world flagging behavior—which is confirmed by our validation. The lesson learned from this parameter is that increasing the tolerance of users would foster moderation, because it reduces the amount of potential conflict.

## 5.3 The effects of integrity ($\gamma$)

We now look at how sources position themselves when they have different opportunities to resist the pressure from flags. For low values of $\gamma$ sources are more swayed, simulating a world in which they are completely subservient to the pressure coming from social media. For high $\gamma$, instead, a news organization has the possibility to resist such a pressure: they choose a specific polarity because they think it is the right world view and only strong interventions can move them.

Integrity seems to have a negligible effect on the polarization of users and sources. Fig 6(a)–6(f) show practically indistinguishable polarity distributions.

However, the social network appears to have smaller clusters of extremist users. The explanation is that the users still exist, but they lost all of their social connections. The reason might be that they are unable to establish new friendships, because they cannot find common sources with people with a similar polarity, due to the small propensity of sources to explore the polarity space.

The lesson learned from this parameter is integrity does not seem to prevent the polarization of users or sources, but it might play a role in not giving extremists a platform to organize on.

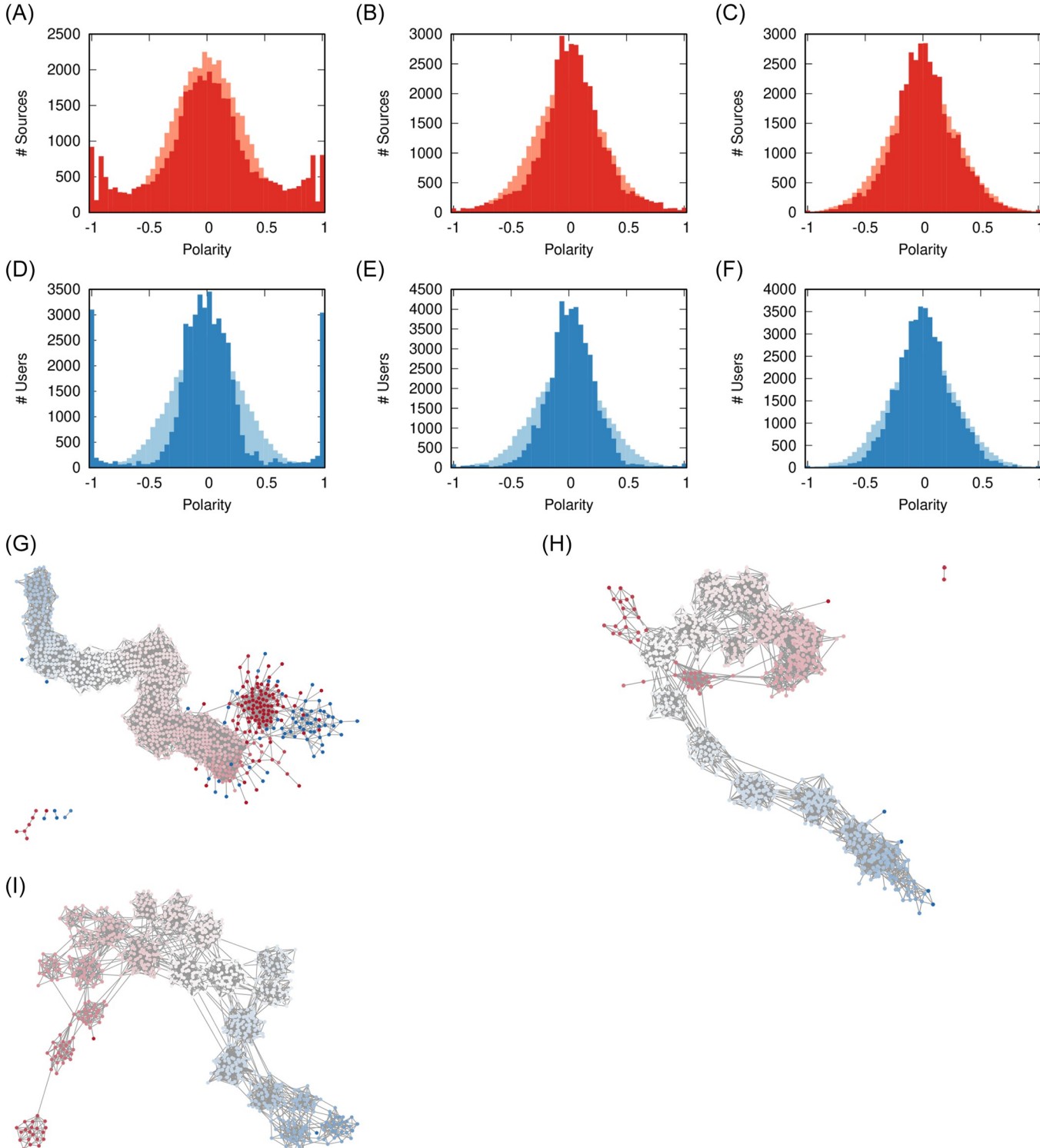

**Fig 5. Effect of $\phi$ on user (a-c) and source (d-f) polarity distributions, and (g-i) on the social network.** The pale distribution in the background is the original one. Other parameters fixed at: $\rho = 0.08$, $\gamma = 0$, $\sigma = 2/3$. The nodes in the network are colored proportionally to their polarity from −1 (blue) to + 1 (red) passing via 0 (white). (a) $\phi = 0.1$ (b) $\phi = 0.2$ (c) $\phi = 0.3$ (d) $\phi = 0.1$ (e) $\phi = 0.2$ (f) $\phi = 0.3$ (g) $\phi = 0.1$ (h) $\phi = 0.2$ (i) $\phi = 0.3$.

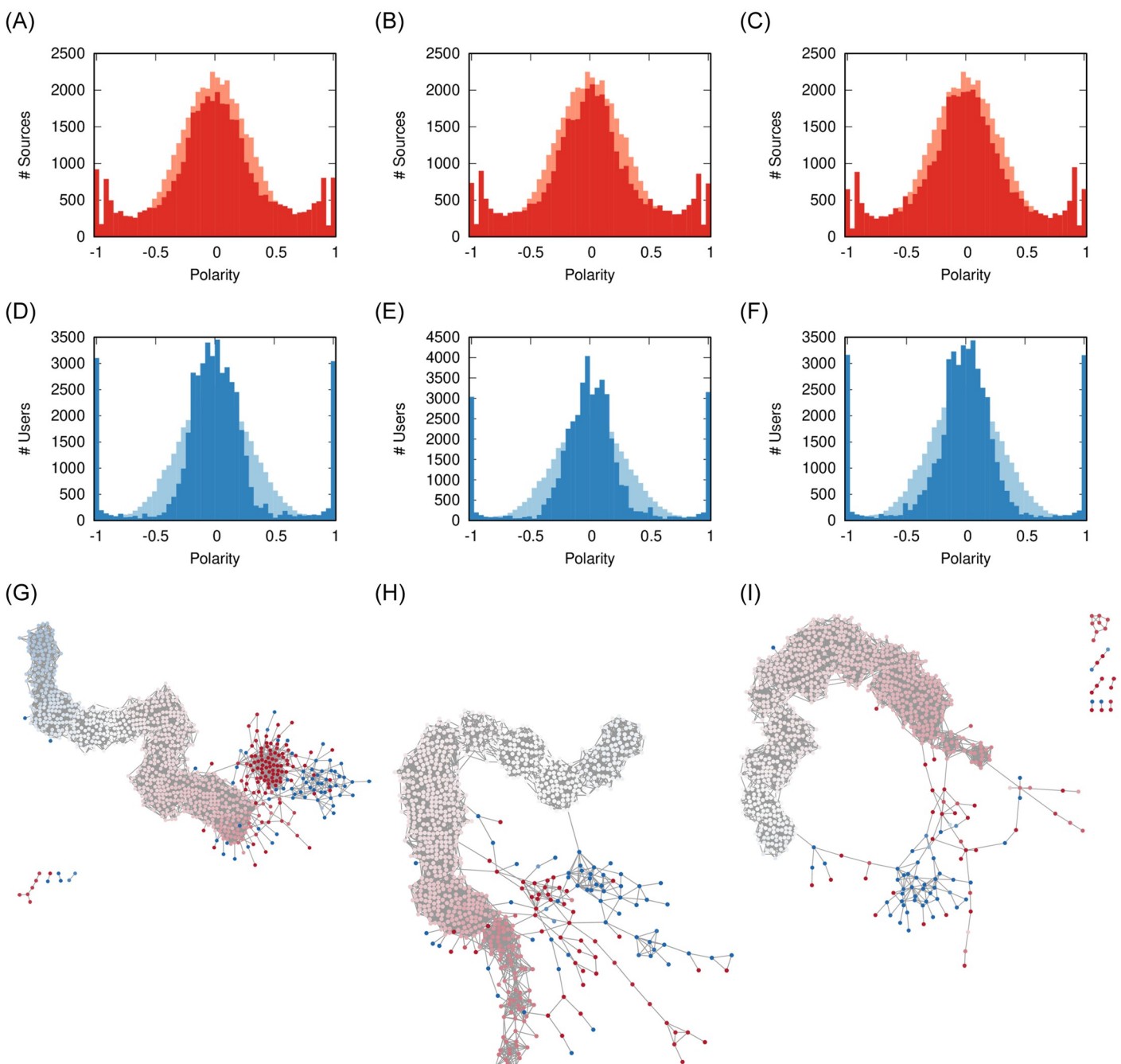

**Fig 6. Effect of $\gamma$ on user (a-c) and source (d-f) polarity distributions, and (g-i) on the social network.** The pale distribution in the background is the original one. Other parameters fixed at: $\rho = 0.08$, $\phi = 0.1$, $\sigma = 2/3$. The nodes in the network are colored proportionally to their polarity from −1 (blue) to +1 (red) passing via 0 (white). (a) $\gamma = 0.0$ (b) $\gamma = 0.1$ (c) $\gamma = 0.2$ (d) $\gamma = 0.0$ (e) $\gamma = 0.1$ (f) $\gamma = 0.2$ (g) $\gamma = 0.0$ (h) $\gamma = 0.1$ (i) $\gamma = 0.2$.

## 5.4 The effects of volatility ($\sigma$)

Finally, we look at the effect of users' opinion volatility. For low values of $\sigma$ users are "stubborn" and value their previous polarity more than what they read on social media. High $\sigma$ implies that users have low critical thinking and move to the average polarity of their friends and the news items they read.

Volatility has a strong effect on the polarization of users. Fig 7(d) shows the effect of maximum volatility: here the cluster of moderates disappears and users must chose a polarity of either + 1 or −1. The resulting social network (Fig 7(g)) loses a majority of its links from unfriending and is entirely composed by two fighting extremist clusters.

When users instead move to the midpoint between their original polarity and the one they feel from social media, there is still source and user polarization (Fig 7(c) and 7(f)). However, the polarized users are removed from the social network (Fig 7(i)).

The lesson learned is that lowering volatility by itself cannot prevent polarization—unless we assume the unrealistic scenario of $\sigma = 0$, i.e. no user ever changes opinion based on their friends and news consumption. The only way to remove polarization is to increase tolerance (Fig 5(f)). However, with smaller volatility—i.e. more user skepticism—the extremists are isolated in the social network. In our validation, we show that real world data is consistent with a high volatility scenario, which would imply high polarization.

## 6 Complex parameter effects

In this section, we investigate complex relationships between pairs of parameters, rather than looking at a single parameter as we did in the previous section. The aim is to show how the model can support a wide array of complex behaviors that are not immediately obvious.

### 6.1 Tolerance & volatility

We start by varying tolerance ($\phi$) and volatility ($\sigma$) over a wider range of values. Fig 8 reports the resulting distributions of user polarity.

The first column (Fig 8(a), 8(d) and 8(g)) replicates the message of Fig 7: lowering volatility lowers polarization. However, looking at the first row (Fig 8(a)–8(c)) we see that we can support low polarization even in presence of high volatility, by increasing tolerance.

This is actually expected due to the mathematical properties of the model: a high-tolerance high-volatility scenario replicates the conditions studied in models of consensus dynamics [39, 40]. With high tolerance, most users are not pushed by anything but only pulled; and with high volatility they will converge quickly towards the average polarity of their neighbors.

### 6.2 Shareability & integrity

We now vary shareability ($\rho$) and integrity ($\gamma$). Fig 9 reports the standard deviations of the source and user polarity distributions. A higher standard deviation implies more dispersed values away from the average and, therefore, more polarity. We use this metric, because the differences in the actual distributions are subtle and not evident when looked at directly.

Both shareability and integrity have a similar effect: increasing them tends to shift the polarization from the sources to the users—the blue line surpasses the red line when moving both left to right and top to bottom in Fig 9.

However, the reasons for this similar effect are different. Shareability increases polarization for both users and sources, but proportionally more for users. Integrity decreases all polarization, but more for sources than for users.

One could speculate that, in the real world, the two parameters are intermingled. A source with higher integrity is less prone to chasing clicks which, in turn, would lower the number of shared news items in the network. Thus, the polarization dampening that happens when increasing integrity (from Fig 6) is stronger than we can see by changing $\gamma$ in isolation, because increasing $\gamma$ indirectly lowers $\rho$ as well.

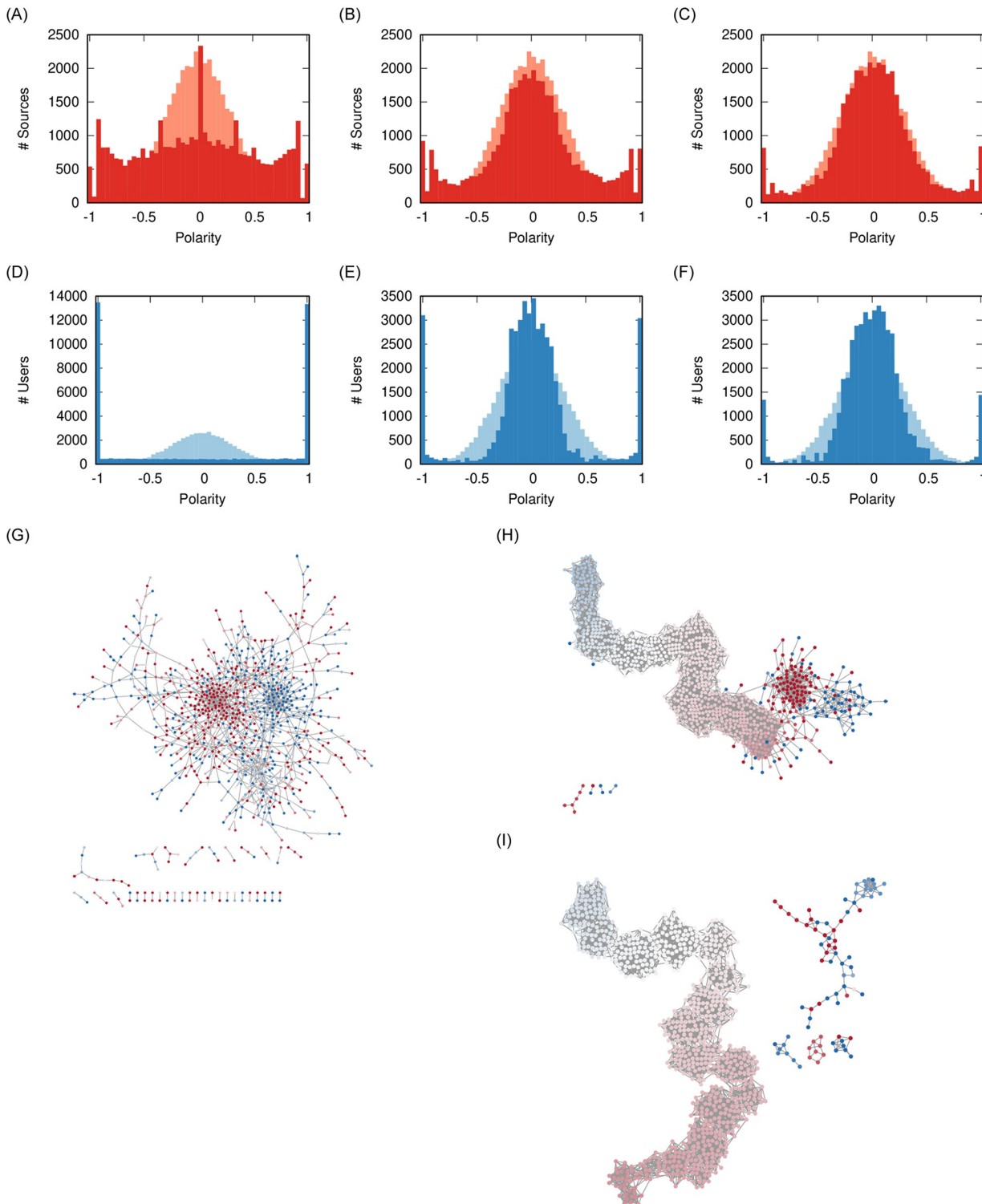

**Fig 7. Effect of σ on user (a-c) and source (d-f) polarity distributions, and (g-i) on the social network.** The pale distribution in the background is the original one. Other parameters fixed at: $\rho = 0.08$, $\phi = 0.1$, $\gamma = 0$. The nodes in the network are colored proportionally to their polarity from −1 (blue) to + 1 (red) passing via 0 (white). (a) $\sigma = 1$ (b) $\sigma = 0.66$ (c) $\sigma = 0.5$ (d) $\sigma = 1$ (e) $\sigma = 0.66$ (f) $\sigma = 0.5$ (g) $\sigma = 1$ (h) $\sigma = 0.66$ (i) $\sigma = 0.5$.

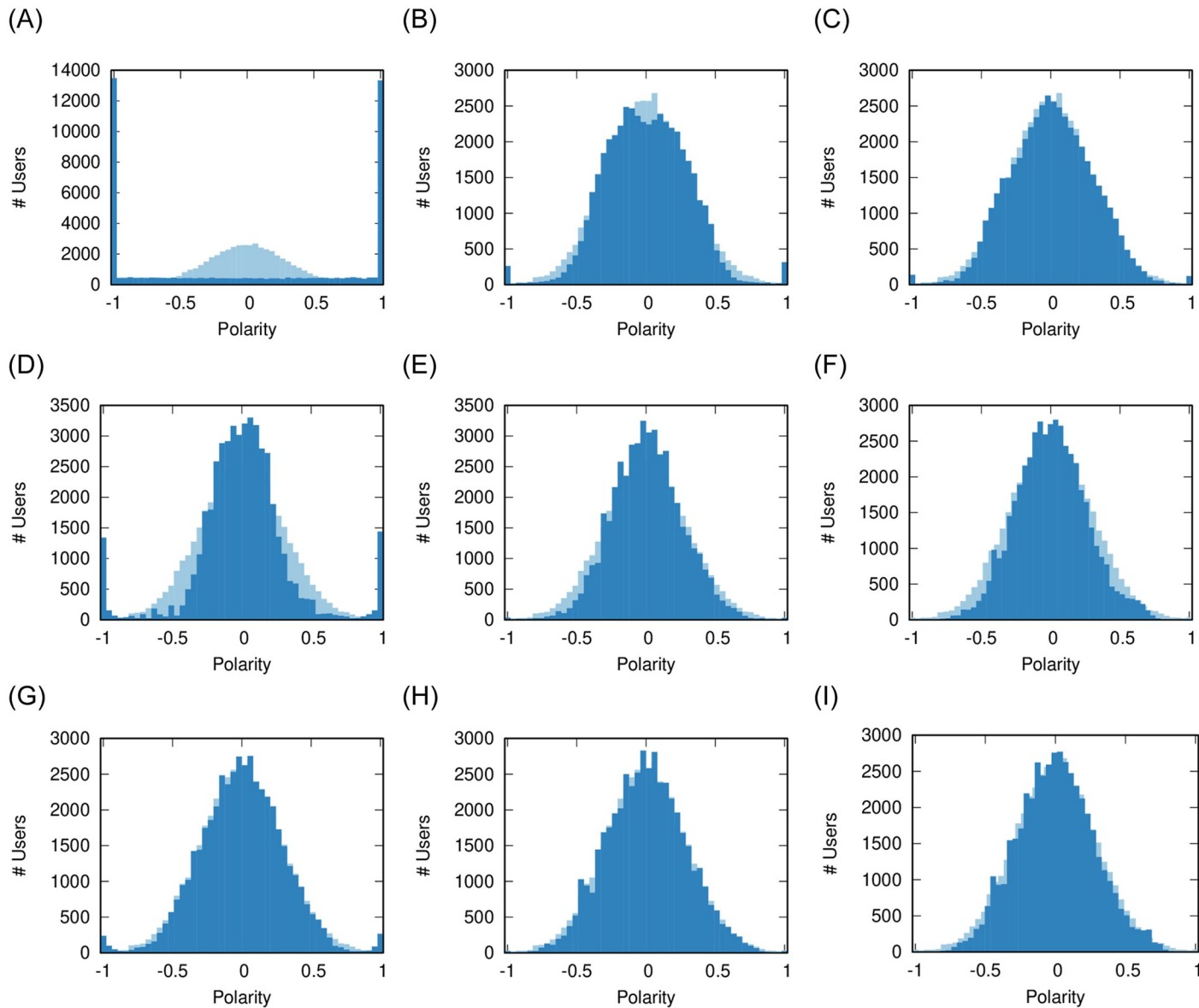

**Fig 8. The user polarity distributions when co-varying φ and σ.** In the figures above, the values of φ are 0.1, 0.3, 0.5 (low, medium, high); the values of σ are 0.1, 0.5, 1 (low, medium, high); while we keep ρ = 0.08 and γ = 0. (a) Low φ − High σ (b) Medium φ − High σ (c) High φ − High σ (d) Low φ − Medium σ (e) Medium φ − Medium σ (f) High φ − Medium σ (g) Low φ − Low σ (h) Medium φ − Low σ (i) High φ − Low σ.

## 6.3 Shareability & tolerance

It is also interesting to study the effect of varying tolerance and shareability, because they are both characteristics of the users. In a sense, they define the playing field on which a news organization needs to play.

Fig 10 depicts the result. Our observations from the previous section are confirmed: between the two parameters, φ has a much larger impact in determining the polarization of the system, as the extremes progressively disappear if we move left to right in the figure—i.e. increasing φ.

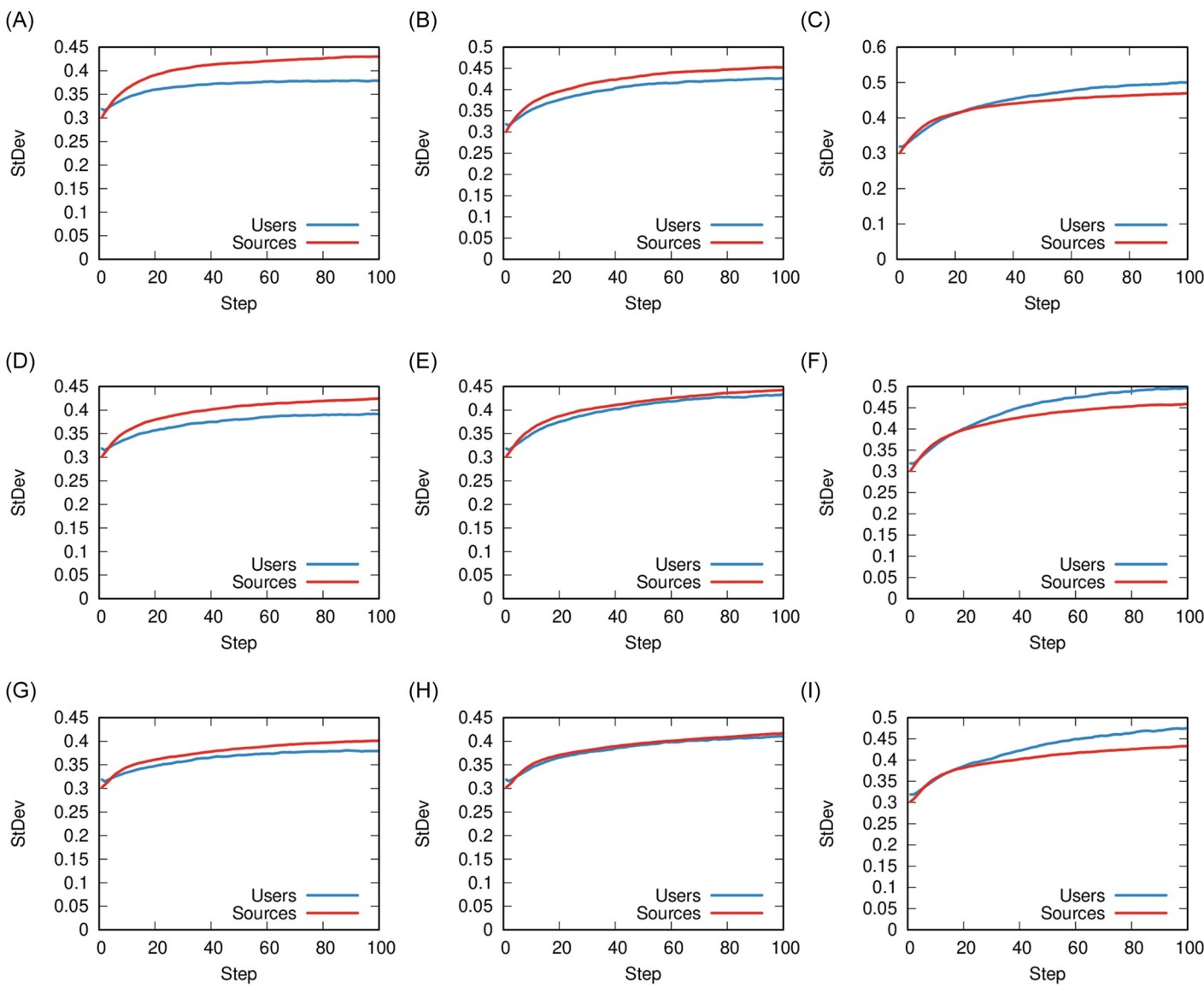

**Fig 9. Standard deviations of the sources (red) and users (blue) polarity distributions when co-varying $\rho$ and $\gamma$.** In the figures above, the values of $\rho$ are 0.06, 0.08, 0.1 (low, medium, high); the values of $\gamma$ are 0, 0.2, 0.4 (low, medium, high); while we keep $\phi = 0.1$ and $\sigma = 0.66$. (a) Low $\rho$ − Low $\gamma$ (b) Medium $\rho$ − Low $\gamma$ (c) High $\rho$ − Low $\gamma$ (d) Low $\rho$ − Medium $\gamma$ (e) Medium $\rho$ − Medium $\gamma$ (f) High $\rho$ − Medium $\gamma$ (g) Low $\rho$ − High $\gamma$ (h) Medium $\rho$ − High $\gamma$ (i) High $\rho$ − High $\gamma$.

However, it is interesting to see how $\rho$ behaves as a sort of amplifier of the main effect of $\phi$. In low tolerant societies (left column) decreasing shareability (moving top to bottom) also decreases polarization. On the other hand, in tolerant societies (middle and right columns), increasing $\rho$ (moving bottom to top) also increases the tightness of the polarity distribution—i.e. the moderates cluster more tightly around zero, perfect neutrality.

## 7 Discussion

Our results gave us several insights into how a complex system where individuals and media sources are regulated by non trivial parameters can evolve into polarization.

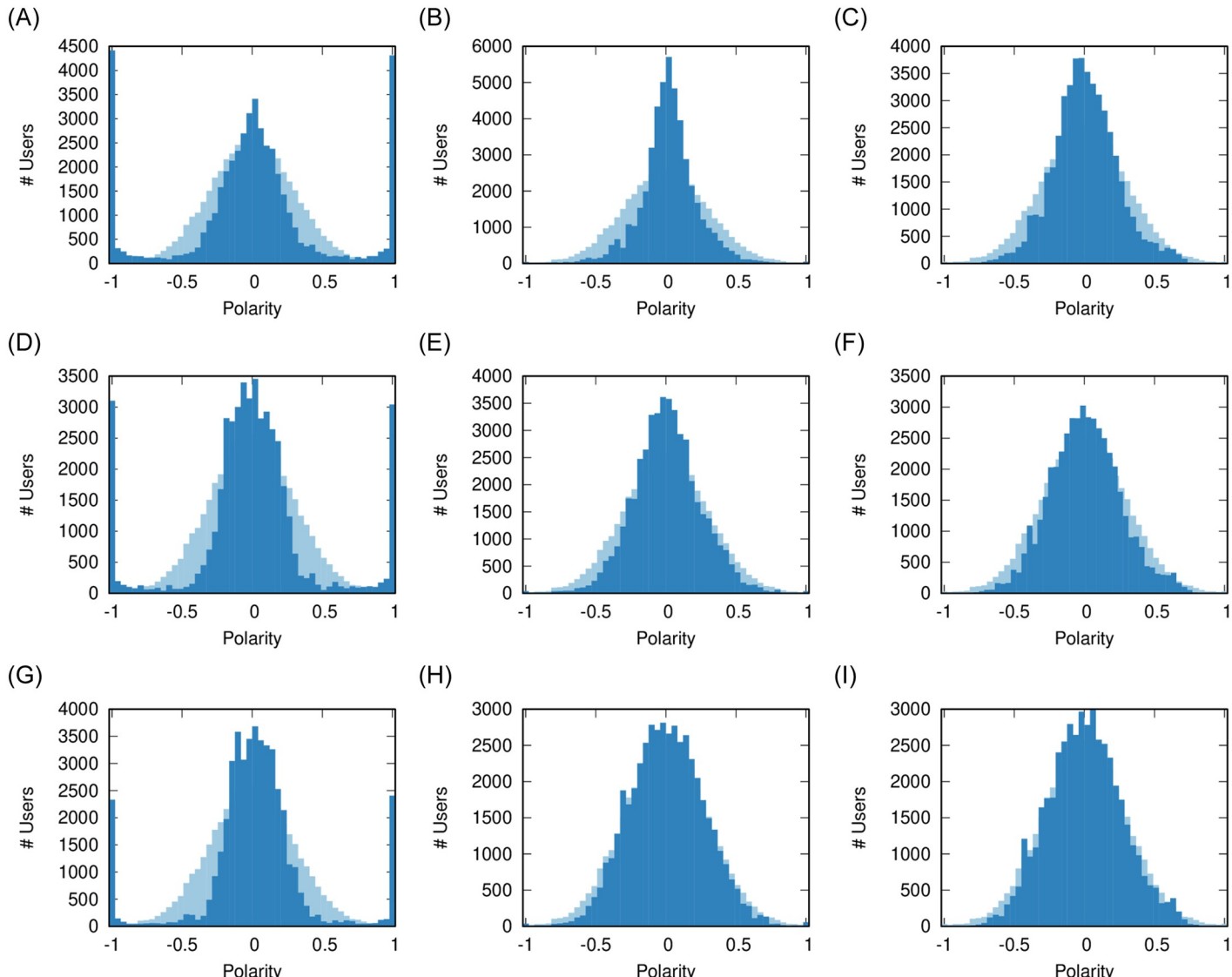

**Fig 10. The user polarity distributions when co-varying ϕ and ρ.** In the figures above, the values of ϕ are 0.1, 0.3, 0.5 (low, medium, high); the values of ρ are 0.06, 0.08, 0.1 (low, medium, high); while we keep σ = 0.66 and γ = 0. (a) Low ϕ − High ρ (b) Medium ϕ − High ρ (c) High ϕ − High ρ (d) Low ϕ − Medium ρ (e) Medium ϕ − Medium ρ (f) High ϕ − Medium ρ (g) Low ϕ − Low ρ (h) Medium ϕ − Low ρ (i) High ϕ − Low ρ.

We have discovered that (i) higher shareability of the news produces higher levels of polarization, (ii) a less tolerant society produces higher level of polarization, (iii) news sources' integrity alone does not affect the overall polarization and (iv) with a higher level of skepticism more extremist users will end up isolated.

Moreover, we have also observed how shreability (ρ), tolerance (ϕ), integrity (σ), and volatility (γ) interact together in non-trivial ways reproducing the scenarios studied in consensus dynamics models—when tolerance and volatility are observed together—and affecting the different actors in the model (users and news sources) in different ways—when shareability and integrity are observed together.

These findings are relevant when we try to put them into societal context. We can use them either to design empirical research that aims at verifying them, or as a basis of policies to mitigate societal polarization.

Shareability has often been seen as a positive attribute for information content. Sources try to achieve higher shareability of their content [41] and social media platforms equate reshares with engagement, which is what constitutes the monetary incentive coming from advertisers [42]. However, recently social media platforms have adopted solutions to reduce the circulation of problematic and potentially polarizing content [43, 44]. Our research suggests that this is a step towards the right direction.

Users' tolerance and skepticism are harder to control on the platforms' side. A growing body of empirical evidence [45] agrees with the importance our model gives to tolerance. The role played by skepticism (volatility, in our model) is harder to grasp. Empirical evidences [46] show how skepticism is frequently associated with pre-existing confirmation biases. A study nudging users to be skeptical [47] was met with success: users tend to be more critical when platforms or media sources suggest to fact check specific news.

A final remark can be done looking at the complex interaction that takes place between different parameters. As we have seen in Figs 8 and 9, multiple parameters produce non trivial effects on the system. This calls for more, so far unavailable, empirical research that takes these more complex dynamics into account.

The archive containing the data and code necessary for the replication of our results can be found at https://www.dropbox.com/s/rldphdm8w6letox/20211020_flagging_code.zip?dl=0.

## Supporting information

**S1 File. Supplementary information.** All the supporting figures.
(PDF)

## Acknowledgments

The authors wish to thank Marilena Hohmann for her help in collecting the data from Twitter.

## Author Contributions

**Conceptualization:** Michele Coscia, Luca Rossi.

**Formal analysis:** Michele Coscia, Luca Rossi.

**Investigation:** Michele Coscia, Luca Rossi.

**Methodology:** Michele Coscia, Luca Rossi.

**Project administration:** Michele Coscia, Luca Rossi.

**Software:** Michele Coscia.

**Validation:** Michele Coscia, Luca Rossi.

**Visualization:** Michele Coscia.

**Writing – original draft:** Michele Coscia, Luca Rossi.

**Writing – review & editing:** Michele Coscia, Luca Rossi.

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
