## [Decision Letter · Decision Letter 0]

19 Aug 2021

PONE-D-21-21470

How Minimizing Conflicts Could Lead to Polarization on Social Media: an Agent-Based Model Investigation

PLOS ONE

Dear Dr. Coscia,

Thank you for submitting your manuscript to PLOS ONE. After careful consideration, we feel that it has merit but does not fully meet PLOS ONE’s publication criteria as it currently stands. Therefore, we invite you to submit a revised version of the manuscript that addresses the points raised during the review process.

We look forward to receiving your revised manuscript.

Kind regards,

Kazutoshi Sasahara

Academic Editor

PLOS ONE

Additional Editor Comments (if provided):

Although they find the proposed model interesting, both reviewers think the current manuscript needs more work. Reviewer1 suggested the validation of the model with the real-world data. Reviewer 2 suggested more investigations about relevant models. Please read their comments carefully and properly incorporate.

Reviewers' comments:

Reviewer's Responses to Questions

**Comments to the Author**

1. Is the manuscript technically sound, and do the data support the conclusions?

Reviewer #1: No

Reviewer #2: Yes

2. Has the statistical analysis been performed appropriately and rigorously? 

Reviewer #1: N/A

Reviewer #2: Yes

3. Have the authors made all data underlying the findings in their manuscript fully available?

Reviewer #1: Yes

Reviewer #2: Yes

4. Is the manuscript presented in an intelligible fashion and written in standard English?

Reviewer #1: Yes

Reviewer #2: Yes

5. Review Comments to the Author

Reviewer #1: In this work, the authors propose an agent-based model for studying polarization on social media. It considers two types of agents, namely users and news sources, as well different parameters to change several characteristics of the agents, such as users' volatility and news outlet integrity. The paper is well written and easy to understand.

Although I appreciate the authors' effort, I am sorry to say not convinced about the validity and soundness of this work.

First of all, as the authors wrote, their data are just the outcome of a simulation, except for some data taken by Crowdtangle to model the audience network. I suggest the authors compare their results with data from the real world to show that the model can mimic reality for some values of the parameters.

Another issue I found is the differences between their results and other works working with real-world data. For example, results on users' polarization look different from the one obtained in some works, e.g Cinelli, Matteo, et al. "The echo chamber effect on social media." Proceedings of the National Academy of Sciences 118.9 (2021).

This difference is quite high in all scenarios making unreliable all the speculation made by the authors.

Moreover, the authors made some assumptions on the starting condition of the system without providing any explanation. One I do not agree with is the initial distribution of the polarity of users and news sources. Why they should be uniformly distributed? I have never seen such a case.

Again, without any comparison with real-world data, this model seems to be purely theoretic, and thus the conclusion the authors made could be not valid for the real world.

Some minor remarks:

-authors wrote that the audience network is bipartite, but it is not represented as bipartite in figure 2a. I suggest the authors change this representation to underline the different nature of this network with respect to the social network.

-authors provided results for a few values, and said that the model does not have significant changes outside the parameters reported. I would like to have an explanation of this behavior, especially because for some parameters (e.g. Reshareability) the range of values is very narrow.

Reviewer #2: The authors present an agent-based model to study the complex interplay between users and news sources and how these interactions lead to polarization of both users and sources. This topic, social polarization, has received substantial attention in recent years and is of vital importance in many different fields. Overall, the paper is well-written with solid motivations and clearly exposed methods. The model hypothesis, based on the previous experimental studies, is reasonable. Depending on the values of the parameters, the model presents some interesting behaviors that might be important in real-world application, for example, higher shareability of policing content may produce higher levels of polarization. Therefore, I think it should be published.

However, I have some criticisms and suggestions on the current manuscript which should be addressed first:

1. I have the feeling that the authors have an intimate knowledge of experimental and empirical studies while are not familiar with the relevant modeling literature. I suggest the authors add some reviews of the previous works on agent-based models of social polarization in Introduction to better highlight their innovations and original contributions, for instance:

The following work also considered the coupling evolutions and the polarization of both users and news outlets:

[1] AL Schmidt et al. "Anatomy of news consumption on Facebook." Proceedings of the National Academy of Sciences 114.12 (2017): 3035-3039.

Some recent advances on social/political polarization using agent-based models:

[2] X Wang et al. "Public discourse and social network echo chambers driven by socio-cognitive biases." Physical Review X 10.4 (2020): 041042.

[3] K Sasahara et al. "Social influence and unfollowing accelerate the emergence of echo chambers." Journal of Computational Social Science 4.1 (2021): 381-402.

Some notable physical models:

[4] F Baumann et al. "Modeling echo chambers and polarization dynamics in social networks." Physical Review Letters 124.4 (2020): 048301.

[5] M Del Vicario et al. "Modeling confirmation bias and polarization." Scientific reports 7.1 (2017): 1-9.

2. One of the main innovations of this paper is the incorporation of homophily on initial conditions (i.e., more realistic starting conditions of user opinions). However, the authors didn’t provide their detailed algorithm for allocating initial polarity values on both audience networks and social networks. Note that this directly affects the current conclusions and should be indispensable.

3. Some technical problems:

(1) The updating rules for polarity seems like cannot always guarantee p_{u,t} in the range of [-1, 1]. Consider an extreme circumstance: \\sigma=1, tolerance \\phi=0.8, p_{u,t-1}=-0.8 and this user is affected by two users with polarity -0.4 (pull) and 0.6 (push), respectively. Then d_{u,t-1}=-0.5, which means p_{u,t}=-1.3.

(2) The termination condition of the dynamical system that guarantees convergence should be provided.

(3) Following (2), I think the authors should add some snapshots of time evolutions of the system, both polarity distribution evolution and social network evolution, to illustrate the emerging process of polarization.

4. I’m curious that if the integrity of news sources depends on both the number of resharing and the number of flags (for instance, the difference between these two numbers), would results in Section 4.3 be qualitatively different?

5. The combined effects of tolerance and shareability (both are users’ properties) should be studied in Section 5.

Some minor comments:

1. Some basic network information used in the simulations should be given. For example, the number of sources and users, the number of edges or the average degree in audience networks and social networks.

2. Line 336: Figure 6(d) show --- Figure 7(d) shows

3. Line 338: Figure 7(h) --- Figure 7(g)

6. PLOS authors have the option to publish the peer review history of their article (what does this mean?). If published, this will include your full peer review and any attached files.

Reviewer #1: No

Reviewer #2: **Yes: **Xin Wang

---

## [Author Response · Author response to Decision Letter 0]

22 Oct 2021

To the editors of Plos One,

thank you for the consideration you are giving to our paper “How Minimizing Conflicts Could Lead to Polarization on Social Media: an Agent-Based Model Investigation”. We also thank the reviewers for their deep comments about the paper. We have done our best to address their points, and we think the paper has greatly improved as a result of this. We hope that you will agree with us.

We follow with a point-by-point response to the reviewers, where we detail either what we did on the paper to take into account their comments or a response to the point. For yours and the reviewers’ convenient, we have submitted a track change version of the paper so that all our edits are easier to locate.

Reviewer #1

 1. Data are just the outcome of a simulation, except for some data taken by Crowdtangle to model the audience network. I suggest the authors compare their results with data from the real world to show that the model can mimic reality for some values of the parameters.

 ◦ We agree with this point. We have collected Twitter data from the same source used by the paper Reviewer #1 suggests in the next point. We have added Section 4 to the paper showing how for some parameter combinations our model does a reasonable job in reproducing that data.

 2. Another issue I found is the differences between their results and other works working with real-world data. For example, results on users' polarization look different from the one obtained in some works, e.g Cinelli, Matteo, et al. "The echo chamber effect on social media." Proceedings of the National Academy of Sciences 118.9 (2021). This difference is quite high in all scenarios making unreliable all the speculation made by the authors.

 ◦ We agree that compatibility of results with existing research is extremely important for works based on simulation such the one we are presenting. Nevertheless, we fail to understand in which way the results we obtain on users’ polarization would look different from what Cinelli et al. obtained. As we show in the newly added Section 4 the model can produce levels of polarization similar to what was reported in Cinelli at al. for Twitter (we used part of the same Twitter network and the same procedure to estimate users’ leaning) . Moreover our model and the analysis in the above-mentioned paper focus on different aspects and results are hard to compare. Even when the two works deal with somehow similar issues (e.g. in the case of spreading) there are substantial differences in how those are implemented (our model builds on a bipartite structure and does not model the actual information propagation through a SIR approach but rather uses information propagation to observe the evolution of users’ opinions) to suggest caution when comparing the results.

 3. The authors made some assumptions on the starting condition of the system without providing any explanation. One I do not agree with is the initial distribution of the polarity of users and news sources. Why they should be uniformly distributed?

 ◦ We agree that a uniform distribution is not realistic. We point out that we use a normal distribution, which is instead supported by empirical evidence, as we show in Section 1 of the Supplementary Material.

 4. Authors wrote that the audience network is bipartite, but it is not represented as bipartite in figure 2a. 

 ◦ We have tried to improve Figure 2a the best way we can, by representing sources as triangles. We hope it is now clearer, but it is indeed difficult to get a detailed view of it given its number of nodes/edges.

 5. Authors provided results for a few values, and said that the model does not have significant changes outside the parameters reported. I would like to have an explanation of this behavior.

 ◦ Excellent point. We provide some support to our choices in Section 2 of the Supplementary Material. Specifically, in the section we show the repercussions of increasing some parameter values (e.g. for a tolerance higher than 0.5, almost no flags are generated).

Reviewer #2

 1. I have the feeling that the authors have an intimate knowledge of experimental and empirical studies while are not familiar with the relevant modeling literature. I suggest the authors add some reviews of the previous works on agent-based models of social polarization […]

 ◦ Thank you for pointing this out. We have added and discussed the suggested references throughout the paper both to acknowledge their contribution to the field as well as to highlight where our model is different and in which way. 

 2. One of the main innovations of this paper is the incorporation of homophily on initial conditions (i.e., more realistic starting conditions of user opinions). However, the authors didn’t provide their detailed algorithm for allocating initial polarity values on both audience networks and social networks. 

 ◦ Excellent point. We do so now in Section 3.3. Specifically, what we do is to assign to users grouped in the same community a contiguous portion of the polarity spectrum.

 3. The updating rules for polarity seems like cannot always guarantee p_{u,t} in the range of [-1, 1]

 ◦ This is entirely correct. We added and explanation about how we guarantee to respect the system’s bound in Section 3.4.1. Specifically, whenever a user or a source goes out of bounds, we cap the value to either +1 or -1.

 4. The termination condition of the dynamical system that guarantees convergence should be provided.

 ◦ We cannot provide an analytic proof of termination, but we provide in Section 3 of the Supplementary Material some empirical data about the stabilization of the system. We terminate the simulation at the 100th step, which is well past the regime change in which the system does not experience much change any more.

 5. I think the authors should add some snapshots of time evolutions of the system, both polarity distribution evolution and social network evolution, to illustrate the emerging process of polarization

 ◦ This is a good point and we did so in Section 5 of the Supplementary Material.

 6. If the integrity of news sources depends on both the number of resharing and the number of flags (for instance, the difference between these two numbers), would results in Section 4.3 be qualitatively different?

 ◦ This is a fair question and the results change, in an almost unnoticeable way. We support this claim in Section 4 of the Supplementary Material. Since this modification would need the introduction of a fifth parameter, we think that the size of the effect is not enough to justify the increase in complexity in the model.

 7. The combined effects of tolerance and shareability (both are users’ properties) should be studied in Section 5.

 ◦ We do now in the new Section 6.3 -- as Section 5 became Section 6.

 8. Some basic network information used in the simulations should be given. For example, the number of sources and users, the number of edges or the average degree in audience networks and social networks.

 ◦ We added this information in the text.

 9. Line 336: Figure 6(d) show --- Figure 7(d) shows [...] Line 338: Figure 7(h) --- Figure 7(g)

 ◦ These minor reference issues are now corrected.

---

## [Decision Letter · Decision Letter 1]

26 Nov 2021

PONE-D-21-21470R1How Minimizing Conflicts Could Lead to Polarization on Social Media: an Agent-Based Model InvestigationPLOS ONE

Dear Dr. Coscia,

Thank you for submitting your manuscript to PLOS ONE. After careful consideration, we feel that it has merit but does not fully meet PLOS ONE’s publication criteria as it currently stands. Therefore, we invite you to submit a revised version of the manuscript that addresses the points raised during the review process.

We look forward to receiving your revised manuscript.

Kind regards,

Kazutoshi Sasahara

Academic Editor

PLOS ONE

Additional Editor Comments:

Although both reviewers think the manuscript was improved, the reviewer 1 is expressing concerns (see comments). In addition, the reviewer 2 suggested two papers.

Reviewers' comments:

Reviewer's Responses to Questions

**Comments to the Author**

1. If the authors have adequately addressed your comments raised in a previous round of review and you feel that this manuscript is now acceptable for publication, you may indicate that here to bypass the “Comments to the Author” section, enter your conflict of interest statement in the “Confidential to Editor” section, and submit your "Accept" recommendation.

Reviewer #1: (No Response)

Reviewer #2: All comments have been addressed

2. Is the manuscript technically sound, and do the data support the conclusions?

Reviewer #1: Partly

Reviewer #2: Yes

3. Has the statistical analysis been performed appropriately and rigorously? 

Reviewer #1: No

Reviewer #2: Yes

4. Have the authors made all data underlying the findings in their manuscript fully available?

Reviewer #1: Yes

Reviewer #2: Yes

5. Is the manuscript presented in an intelligible fashion and written in standard English?

Reviewer #1: Yes

Reviewer #2: Yes

6. Review Comments to the Author

Reviewer #1: Although I found the paper improved in many aspects, and I recognize the effort spent by the authors to answer my requests, I still have some concerns about this work.

I appreciate that the authors added Twitter data in their analysis, but I have some concerns about the conclusions they drew from them.

The authors claim they can reasonably reproduce the results for a dataset from the reference I suggested and used Wasserstein distance to confirm this result. However, I have still hold doubts regarding the users' distributions.

Figure S7 shows the distribution obtained by the model for the parameters that best fit Twitter data.

However, the distribution is quite different from the marginal one shown in figure 1a of the reference I mentioned. Indeed figure S7 of this paper show a high peak at the center that is not present in the top marginal distribution figure 1a of the suggested reference.

Nevertheless, the authors did not show the distribution of users polarization obtained by the Twitter data they retrieved, and thus I cannot say whether or not their model can effectively reproduce it in case it has significant differences from the one shown in the reference.

I agree with the authors that the results on the spreading dynamics are not directly comparable, but I think that the user polarization distributions are comparable, and figure S7 did not convince me the model can reproduce real data.

More examples of polarization distribution from social data can be found in Cota, Wesley, et al. "Quantifying echo chamber effects in information spreading over political communication networks." EPJ Data Science 8.1 (2019): 1-13, in Zollo, Fabiana, et al. "Debunking in a world of tribes." PloS one 12.7 (2017): e0181821 and in Flamino, James, et al. "Shifting Polarization and Twitter News Influencers between two US Presidential Elections." arXiv preprint arXiv:2111.02505 (2021).

Hence, I suggest the authors show the real data distribution.

Moreover, I think the paper is missing a brief discussion on the parameters that best fit real data to help the reader understand which characteristics we need to fit Twitter data (e.g. high/low sharing, tolerance, etc.)

Reviewer #2: The authors have taken significant care of the details brought to attention by the reviewer comments, and I’m satisfied with all of their responses. In particular, the revised manuscript has added some important technical details, which significantly improve the readability and scientific rigor. As I mentioned in the previous report, the dynamical mechanism that raises the emergence of social polarization has received substantial attention in recent years, and the current work is a timely contribution to this opening problem. Therefore, I recommend accepting it for publication in PLOS ONE.

P.S: I’ve noticed that the authors still missed two important references (which were also mentioned in my previous report) that addressed the coupling evolutions and the polarization of both users and news outlets using agent-based models, and I suggest the authors include them in the introductory discussion:

[1] AL Schmidt et al. "Anatomy of news consumption on Facebook." Proceedings of the National Academy of Sciences 114.12 (2017): 3035-3039.

[2] X Wang et al. "Public discourse and social network echo chambers driven by socio-cognitive biases." Physical Review X 10.4 (2020): 041042.

7. PLOS authors have the option to publish the peer review history of their article (what does this mean?). If published, this will include your full peer review and any attached files.

Reviewer #1: No

Reviewer #2: No

---

## [Author Response · Author response to Decision Letter 1]

14 Dec 2021

Reviewer #1

1) The authors claim they can reasonably reproduce the results for a dataset from the reference I suggested and used Wasserstein distance to confirm this result. However, I have still hold doubts regarding the users' distributions. Figure S7 shows the distribution obtained by the model for the parameters that best fit Twitter data. However, the distribution is quite different from the marginal one shown in figure 1a of the reference I mentioned. Indeed figure S7 of this paper show a high peak at the center that is not present in the top marginal distribution figure 1a of the suggested reference.

A) Part of the reason why Fig S7 did not look like Fig 1a in the reference was that we worked with the "guncontrol" data and the reference worked with the "abortion" data. We have now switched to using the "abortion" data to make the comparison possible. However, the main reason for the difference rested on the fact that our validation relied on an unmet assumption. Specifically, we assumed that mediabiasfactcheck allowed sources to take polarity values in the full -1/+1 polarity spectrum. This is not true: no source in mediabiasfactcheck has a polarity of either -1 or +1, making those polarities unattainable for users. Thus, this makes a direct comparison with our model imperfect, because our ABM allows users to take values in the full -1/+1 polarity spectrum -- which is routinely done in the literature (e.g. Wang et al, Physical Review X, 2020). We have now normalized the mediabiasfactcheck data to have the same value domain as our model, under the assumption that we are aligning our theoretical maximum/minimum values with the observed maximum/minimum values of polarization. We also used a more intuitive test relying on the Pearson correlation coefficient in Sections 4 and S5. As can be seen comparing Fig S7(right) with the real world distribution (Fig S8 (left)), the two distributions exhibit similar features, with two large peaks at either side of neutrality and little in between, if Fig S8's x axis were to be normalized. Moreover, even the topology of the model's network (Fig S9 (bottom right)) shows the main two-community feature of the real world data (Fig S8 (right)).

2) Nevertheless, the authors did not show the distribution of users polarization obtained by the Twitter data they retrieved, and thus I cannot say whether or not their model can effectively reproduce it in case it has significant differences from the one shown in the reference. (...) Hence, I suggest the authors show the real data distribution.

A) We now show the real world data in Fig S8, both polarity distribution and network topology.

3) I think the paper is missing a brief discussion on the parameters that best fit real data to help the reader understand which characteristics we need to fit Twitter data (e.g. high/low sharing, tolerance, etc.)

A) We have now added this as the conclusion of Section 4. Specifically we show that high sharing, low tolerance, and high volatility reproduce well the real world data, while integrity seems not to be playing a role.

Reviewer #2

1) I’ve noticed that the authors still missed two important references (which were also mentioned in my previous report) that addressed the coupling evolutions and the polarization of both users and news outlets using agent-based models, and I suggest the authors include them in the introductory discussion (...)

A) We apologize for this oversight and we have now included these two papers in our introductory discussion. They are now Ref 6 (Wang et al) and Ref 10 (Schmidt et al).

---

## [Decision Letter · Decision Letter 2]

14 Jan 2022

How Minimizing Conflicts Could Lead to Polarization on Social Media: an Agent-Based Model Investigation

PONE-D-21-21470R2

Dear Dr. Coscia,

We’re pleased to inform you that your manuscript has been judged scientifically suitable for publication and will be formally accepted for publication once it meets all outstanding technical requirements.

Kind regards,

Kazutoshi Sasahara

Academic Editor

PLOS ONE

Additional Editor Comments (optional):

Thank you for revising carefully. Now both reviewers think that all the comments were properly addressed.

Reviewers' comments:

Reviewer's Responses to Questions

**Comments to the Author**

1. If the authors have adequately addressed your comments raised in a previous round of review and you feel that this manuscript is now acceptable for publication, you may indicate that here to bypass the “Comments to the Author” section, enter your conflict of interest statement in the “Confidential to Editor” section, and submit your "Accept" recommendation.

Reviewer #1: All comments have been addressed

Reviewer #2: All comments have been addressed

2. Is the manuscript technically sound, and do the data support the conclusions?

Reviewer #1: Yes

Reviewer #2: Yes

3. Has the statistical analysis been performed appropriately and rigorously? 

Reviewer #1: Yes

Reviewer #2: Yes

4. Have the authors made all data underlying the findings in their manuscript fully available?

Reviewer #1: Yes

Reviewer #2: Yes

5. Is the manuscript presented in an intelligible fashion and written in standard English?

Reviewer #1: Yes

Reviewer #2: Yes

6. Review Comments to the Author

Reviewer #1: (No Response)

Reviewer #2: The authors have addressed all my concerns and I haven no additional suggestions. I recommend accepting it for publication in PLOS ONE.

7. PLOS authors have the option to publish the peer review history of their article (what does this mean?). If published, this will include your full peer review and any attached files.

Reviewer #1: No

Reviewer #2: No

---

## [Editor Report · Acceptance letter]

18 Jan 2022

PONE-D-21-21470R2 

How Minimizing Conflicts Could Lead to Polarization on Social Media: an Agent-Based Model Investigation 

Dear Dr. Coscia:

I'm pleased to inform you that your manuscript has been deemed suitable for publication in PLOS ONE. Congratulations! Your manuscript is now with our production department. 

Kind regards, 

on behalf of

Dr. Kazutoshi Sasahara 

Academic Editor

PLOS ONE